# Physiological Function during Exercise and Environmental Stress in Humans—An Integrative View of Body Systems and Homeostasis

**DOI:** 10.3390/cells11030383

**Published:** 2022-01-24

**Authors:** Gavin Travers, Pascale Kippelen, Steven J. Trangmar, José González-Alonso

**Affiliations:** 1The European Astronaut Centre, The European Space Agency, Linder Höhe, 51147 Cologne, Germany; gavin.travers@esa.int; 2Centre for Human Performance, Exercise and Rehabilitation, Brunel University London, Uxbridge UB8 3PH, UK; pascale.kippelen@brunel.ac.uk; 3Division of Sport, Health and Exercise Sciences, Department of Life Sciences, Brunel University London, Uxbridge UB8 3PH, UK; 4School of Life and Health Sciences, University of Roehampton, London SW15 4JD, UK; Steven.trangmar@roehampton.ac.uk

**Keywords:** dehydration, hyperthermia, brain, lungs, heart, muscles

## Abstract

Claude Bernard’s *milieu intérieur* (internal environment) and the associated concept of homeostasis are fundamental to the understanding of the physiological responses to exercise and environmental stress. Maintenance of cellular homeostasis is thought to happen during exercise through the precise matching of cellular energetic demand and supply, and the production and clearance of metabolic by-products. The mind-boggling number of molecular and cellular pathways and the host of tissues and organ systems involved in the processes sustaining locomotion, however, necessitate an integrative examination of the body’s physiological systems. This integrative approach can be used to identify whether function and cellular homeostasis are maintained or compromised during exercise. In this review, we discuss the responses of the human brain, the lungs, the heart, and the skeletal muscles to the varying physiological demands of exercise and environmental stress. Multiple alterations in physiological function and differential homeostatic adjustments occur when people undertake strenuous exercise with and without thermal stress. These adjustments can include: hyperthermia; hyperventilation; cardiovascular strain with restrictions in brain, muscle, skin and visceral organs blood flow; greater reliance on muscle glycogen and cellular metabolism; alterations in neural activity; and, in some conditions, compromised muscle metabolism and aerobic capacity. Oxygen supply to the human brain is also blunted during intense exercise, but global cerebral metabolism and central neural drive are preserved or enhanced. In contrast to the strain seen during severe exercise and environmental stress, a steady state is maintained when humans exercise at intensities and in environmental conditions that require a small fraction of the functional capacity. The impact of exercise and environmental stress upon whole-body functions and homeostasis therefore depends on the functional needs and differs across organ systems.

## 1. Introduction

Elucidation of whole-body and organ systems functions during exercise and environmental stress has a long and rich history of important discoveries that have helped shape our current understanding of the body systems interactions during physical stress and their impact on the maintenance of homeostasis, the topic of this honorary issue in memory of Professor Claude Bernard. Conceptually, maintenance of physiological systems functions and cellular homeostasis (regulation of internal milieu) during physical work hinges upon the precise matching of cellular energetic demand and supply, and the production and clearance of metabolic by-products [1,2]. A long-standing question in integrative physiology is whether this actually happens across all exercise modalities and intensities that humans undertake [3,4,5]. The comprehensive characterisation of the neural, respiratory, cardiovascular, metabolic, thermal, endocrine and autonomic adjustments to short incremental or prolonged strenuous exercise in absence or in combination with environmental stress has proved very insightful in addressing this fundamental question [3,6,7,8,9,10,11,12,13,14,15,16,17,18]. Accordingly, this review will examine how whole-body functions and dynamic homeostasis are affected by exercising at different intensities and durations. The impact of exercise and environmental stress on body functions will be interpreted primarily according to the Fick principle, which determines the rate of oxygen consumption by the human body, an organ, limb or tissue. The rate of oxygen consumption is equal to the product of blood flow and the arterial-venous oxygen content differences (a-vO_2_ diff). To gain insights into the underpinning peripheral and central regulatory mechanisms, we will also discuss factors influencing: (i) the delivery of oxygen, nutrients and regulatory substances to the brain and the respiratory, cardiac and skeletal muscles, and (ii) the removal of metabolic by-products (primarily carbon dioxide (CO_2_) and heat) by the lungs and circulation to the environment. The convective transport of gases, nutrients and regulatory substances to organs and tissues is determined by their blood contents and blood flow [4,19]. The central role of blood flow in the integrative responses to exercise and environmental stress will be highlighted. This review will be divided into four main sections that present the adjustments of the human brain, the lungs, the heart and the skeletal muscles to exercise and environmental stress. An integrative section will provide a synthesis of these organ systems responses to physical and environmental stress, and emphasise the role of brain-body interactions on homeostasis. As most human studies in this field have been carried out in young, trained people, it is important to note that the *absolute* physiological responses discussed in this review differ from those of other populations possessing lower physical fitness levels (e.g., untrained adults, children, elderly people, or clinical patients). Cross-population comparisons are outside the scope of this review, and thus will not be discussed. Notwithstanding this limitation, the fundamental physiological principles under scrutiny could be generalisable to other groups, as they are based on the *relative and time course* adjustments to exercise and thermal stress. The interested reader is directed to other topical articles and comprehensive reviews focusing on the magnitude and pattern of response in different populations [20,21,22,23].

## 2. Brain Function during Exercise and Environmental Stress

The brain plays an essential role in maintaining whole-body function and homeostasis. Afferent metabolic, thermal, mechanical, and biochemical signals from multiple body systems and peripheral tissues, are integrated by the brain. Within the brain, afferent inputs are processed, and appropriate efferent neural outputs are generated to support regulation of organ systems function. The brain is also concerned with maintaining its own homeostasis to support its energetic demands. At rest, the brain consumes a substantial proportion (~20%) of the whole-body oxygen uptake [24,25]. During exercise, however, this proportion is comparatively modest (1–2% of *V*O_2max_) despite the apparent profound increases in regional motor neuron and cardiorespiratory activity responsible for activation of locomotor muscles and supporting oxygen transport systems [26,27,28]. In this section, we examine how regional and global blood flow, metabolism, and neural activity of the brain are altered when exercise is performed, with and without added environmental stress.

### 2.1. Exercise Responses

At rest, global cerebral blood flow (gCBF) is ~750 mL·min^−1^, or ~15% of the resting cardiac output (*Q*) [29]. CBF responds in a biphasic manner during incremental exercise. From rest to low-to-moderate intensity exercise (up to ~60% of *V*O_2max_), CBF increases by ~15–35% (~115–250 mL·min^−1^). This increase in CBF is reflected when measured regionally (rCBF) by radioactive tracers or radiolabelled microspheres [26,30], or transcranial Doppler [31,32,33], and when measured globally through summation of volumetric measures of carotid and vertebral artery blood flow [34]. When incremental exercise progresses to the heavy and severe intensity domains (≥60% of *V*O_2max_), CBF begins to fall towards baseline values [32,33,34,35,36]. The reduction in CBF could curtail the cerebral *V*O_2_, which in turn might impair brain function. However, concomitant to the fall in CBF, the cerebral a-vO_2_ diff increases (to the order of ~40% compared to submaximal exercise) and, as a result, the cerebral *V*O_2_ either remains stable [36] or slightly increases [33,37]. Notwithstanding the changes in CBF and a-vO_2_ diff, the cerebral *V*O_2_ for the whole brain is equivalent to only ~0.08 L·min^−1^, representing less than 2% of systemic *V*O_2_ during near-maximal whole-body exercise.

In contrast to the now well-characterised CBF responses to incremental exercise, there is comparatively less known about the CBF response to prolonged, and very-prolonged submaximal exercise in the low-to-moderate exercise intensity domain in normothermic conditions. Of the limited data available, CBF is demonstrated to be elevated by ~20% from rest to exercise, during work performed at a fixed intensity [38], and during self-paced exercise [39]. If there are no further homeostatic disturbances and, in particular, if blood gas homeostasis is maintained at a similar level throughout prolonged exercise (discussed further in Section 3) CBF remains stable [38]. On the other hand, during time-trial exercise and when there is some degree of cardiovascular strain, CBF might fall slightly, with this fall being smaller compared to high or even severe-intensity exercise [39]. When CBF remains stable, this is also reflected in the brain metabolism, with the brain a-vO_2_ diff, cerebral *V*O_2_ and brain glucose uptake staying unchanged [40].

### 2.2. Impact of Adding Thermal Stress to Exercise

When incremental exercise is performed in a compensable hot environment, there is a ~20–25% rise in CBF from rest to submaximal intensities, as seen during similar exercise in thermoneutral conditions [36]. In hotter, uncompensable conditions CBF still increases with increasing exercise intensity but the extent of the rise at any given work rate is reduced [35,41]. As in thermoneutral conditions, the rise in CBF at submaximal exercise intensities is accompanied by a fall in brain a-vO_2_ diff, resulting in the maintenance of cerebral *V*O_2_ to baseline levels [36]. Furthermore, when incremental exercise in a hot environment progresses beyond ~50–60% of maximal work rate, CBF is blunted concomitant to a rise in core body temperature (T_c_) [35,36]. When constant-load cycling exercise is performed at high intensity in a hot environment, a reduction in CBF occurs close to volitional fatigue [42].

During prolonged, constant-load submaximal exercise in a hot environment, the rest-to-exercise transition results in a similar ~20% increase in CBF to that seen during the same exercise performed in normothermic conditions. Thereafter, however, CBF falls back towards, or even below baseline levels (which contrasts to normothermic conditions where CBF is maintained above baseline). In an uncompensable hot environment (40 °C, 20% relative humidity), the development of significant hyperthermia (with an end exercise T_c_ ~40 °C) lowers CBF by ~25% [38]. In a compensable hot environment (35 °C, 30% relative humidity), where end-exercise T_c_ is ~38.5–39 °C, the fall in CBF is still significant (~15%) [43]. Furthermore, the superimposition of dehydration in the same environmental conditions lowers CBF further still [43,44]. CBF is also reduced for much of the duration of self-paced exercise in the heat compared to exercise in a normothermic environment [39]. Despite the suppression of CBF during prolonged, submaximal exercise in environmental stress, the cerebral *V*O_2_ has been shown to be broadly stable, owing to an increase in cerebral O_2_ extraction [43].

### 2.3. The Control of Brain Blood Flow during Exercise, with and without Environmental Stress

The functional significance of the biphasic changes in CBF during incremental exercise, the reduction in CBF with prolonged exercise with thermal stress, and the precise mechanisms regulating these responses remain to be fully elucidated. Within the brain, localised elevations in CBF are thought to reflect both neural feedforward [45,46] and metabolic feedback mechanisms [47,48] in response to increased neuronal activity and metabolic demand (“neurovascular coupling”). This neural-metabolic flow coupling is observed during visual stimulation [49,50,51] and the performance of motor tasks (e.g., hand movements) engaging a small fraction of the total muscle mass [30]. Whether neural activity per se is a major factor in increasing gCBF during dynamic whole-body exercise remains unclear. However, in support of this idea, elevations in rCBF to regions of the brain responsible for cardiorespiratory and gross motor control have been observed in miniature swine [26]. Moreover, O_2_ supply and oxygenation in the frontal cortex (a region of the brain of importance for gross motor control) has been reported to increase with increasing exercise intensity (at least, up to the boundary between moderate and heavy exercise domains) [52,53,54].

Numerous mechanisms are purported to play a role in regulating cerebrovascular tone and regional CBF at the onset of dynamic whole-body exercise [55,56]. A small increase in the partial pressure of CO_2_ in arterial blood (PaCO_2_) is seen at the commencement of dynamic exercise, which would be expected to lead to cerebral vasodilation and explain the increase in rCBV in the middle cerebral artery [57]. For the posterior circulation, where rCBF increases progressively with incremental exercise intensity [34], a temperature-dependent mechanism similar to that operating in the extracranial circulation [34,35,36] has recently been suggested [58]. Finally, the wide-spread vasodilator agent, nitric oxide (NO), could play a role in neurovascular coupling [47]. Of the potential mechanisms underpinning the fall in rCBF and gCBF during high-to-maximal intensity exercise, the hyperventilation-induced reduction in PaCO_2_ (hypocapnia) seems to be of primary importance (discussed further in Section 3). The hyperventilation and associated hypocapnia seen during strenuous exercise in a thermoneutral environment are heightened when the same exercise is performed in environmental stress, explaining why the fall in CBF is amplified in such conditions. In fact, the role of core body hyperthermia, and associated hypocapnia, in the lowering of CBF is supported by studies increasing T_c_ independent of exercise [59,60,61].

Currently, a direct and comprehensive measure of rCBF and oxygenation of the multiple regions of the human brain during incremental exercise is lacking. As discussed in the above sections, and in accordance with the Fick principle, the increase in CBF, at the global level, is coupled to a proportional reduction in the cerebral a-vO_2_ diff, such that the global cerebral *V*O_2_ does not rise (at least at exercise intensities ≤ ~50% of *V*O_2max_) (Figure 1) [33,36,37]. The absence of a change in whole-brain *V*O_2_ suggests that, whilst some regions of the brain increase their metabolic activity during incremental exercise (e.g., motor cortex), the metabolic rate in other regions might be maintained or downregulated. On the other hand, an increase in the cerebral *V*O_2_, in combination with a reduction in CBF during prolonged exercise and severe hyperthermia, is estimated to lower the cerebral mitochondrial oxygen tension (P_mito_O_2_) by ~5 mmHg, which could lower brain oxygenation [62]. However, the extent of the composite reductions in CBF and cerebral oxygenation, even to levels seen at volitional exhaustion (with and without environmental stress) is unlikely to affect brain function [63].

Whilst the combination of strenuous exercise and environmental stress does not appear to compromise cerebral aerobic metabolism, the development of high body temperatures, coupled with reduced CBF, might yet impact on brain homeostasis in other ways. Firstly, the onset of fatigue processes during exercise is at least in part related to the attainment of a high core and brain temperature [66,67,68]. Direct measurements of human brain tissue temperature, or carotid artery blood temperature have not been attempted in exercising humans, but evidence from temperature measures of the blood draining the brain have provided some insight into brain temperature dynamics during exercise in the heat [69]. The temperature of the brain is mainly dependent on its metabolic heat production and the convective heat exchange between the brain and body core [69,70]. During exercise in normothermic conditions, brain temperature increases in association with the rise in T_c_ (and thus arterial blood temperature entering the brain via the carotid and vertebral arteries), whereas CBF and brain metabolic heat production remain stable (as evidenced by constant cerebral *V*O_2_) [36,40,69]. In comparison, when the same exercise is performed in an uncompensable hot environment, a small increase in brain heat production, coupled to a reduction in CBF (~20%) and heat removal (~30%), appear to collectively cause additional heat to be stored within the brain [69]. Nevertheless, the small difference between core and jugular venous blood temperature points towards a more important role of arterial blood temperature in the brain thermal response to exercise and thermal stress compared to increases in brain heat production [69]. Understanding the impact of rising brain temperature and associated exercise stressors on cerebral motoneuron activity and central drive is therefore of paramount importance in elucidating the chain of events leading to fatigue during exercise with and without environmental stress.

Increases in core and brain temperature during exercise are associated with alterations in electroencephalographic (EEG) activity. These alterations are synonymous with the development of fatigue of the central nervous system [38,71,72], reductions in cognitive acuity [73] and decreased arousal [74,75]. The changes in EEG activity appear to be unrelated to reductions in CBF (which is typically seen during exhaustive exercise) [72]. In addition to alterations in cerebrocortical activity, reductions in maximal voluntary force production and voluntary activation, particularly during sustained muscle contractions [38,74], have been observed with passive [74,76,77] and exercise-induced [38,73] elevations in T_c_. This reduction in force generating capacity measured before and after exhaustive exercise is generally considered to be supraspinal (or “central”) in origin, although hyperthermia-related disturbances at the spinal and peripheral level could also interfere with appropriate neuromuscular function [76]. Whilst there is a strong theoretical basis for the role of the central and peripheral nervous system in the fatigue seen during intensive exercise and environmental stress [74,78], it is interesting to note that the locomotor muscle integrated electromyographic (iEMG) activity increases with exercise intensity through to volitional exhaustion [79] and during exhaustive, constant-load maximal aerobic exercise [77,80]. These important observations support that the central nervous system drive to the contracting skeletal muscles increases during strenuous exercise with and without environmental stress.

In summary, intense exercise and thermal stress lead to a reduction in cerebral perfusion. Whilst the fall in O_2_ delivery could conceivably impair cerebral metabolism and brain homeostasis, brain oxygen uptake appears to not be compromised. Reductions in CBF during exercise and environmental stress might, however, contribute to fatigue processes through elevations in brain temperature and alterations in brain function beyond suppression of central drive (a possibility to be further discussed in the integrative Section 6).

## 3. Lung Function during Exercise and Environmental Stress

The respiratory system is a key ‘gatekeeper’ of blood gas homeostasis. Through modulation of the rate and depth of breathing and resulting changes in pulmonary and alveolar ventilations and in alveolar-capillary diffusion, the respiratory system plays a crucial role in the chain of interlinked events that enable adequate supply of O_2_ to body tissues, removal of CO_2_ and maintenance of blood pH. During exercise, as the metabolic demand of the working muscles increases (with more O_2_ extracted from the blood and more CO_2_ and H^+^ released into it), blood homeostasis can be seriously challenged; tight control of breathing is therefore needed.

### 3.1. Exercise Responses

At low-to-moderate exercise intensities, increases in pulmonary ventilation (*V*E) occur instantly when exercise commences (phase 1 of hyperpnea). *V*E then increases more slowly (phase 2) until a steady state is reached (phase 3). *V*E stabilises at a level commensurate to the metabolic needs, thereby maintaining PaCO_2_ and PaO_2_ close to baseline [81]. In the heavy exercise domain (coincident with lactic acidosis), a steady state is never reached. To partly compensate for metabolic acidosis, hyperventilation occurs (i.e., *V*E increases out of proportion to *V*O_2_ and *V*CO_2_). Consequently, at high exercise intensities, PaCO_2_ falls, when PaO_2_ is usually maintained [82].

Some well-conditioned male endurance athletes [83,84] and some females of variable aerobic fitness level [85] are unable to maintain arterial blood gases homeostasis during exercise, showing significant falls in PaO_2_ (>10 mmHg) and arterial oxyhaemoglobin saturation (SaO_2_, >5%) from rest. Insufficient alveolar hyperventilation at submaximal exercise intensities [83,86] and hampered gas exchange (due to ventilation-perfusion ratio maldistribution and diffusion limitation) at higher intensities [87,88] commonly contribute to arterial hypoxemia. Nonetheless, following a reduction in plasma volume and associated haemoconcentration during exercise, arterial O_2_ content (CaO_2_) is preserved or increased, even in those individuals with exercise-induced arterial hypoxemia [84,85]. When exercise is performed with no additional environmental stress, the healthy respiratory system seems therefore generally able to cover the increased O_2_ demand imposed by exercise.

### 3.2. Impact of Adding Environmental Stress to Exercise

Environmental and physiological stress, such as heat, hyperthermia and dehydration, can alter resting lung function independently of exercise [89,90,91]. Further, it has long been established that humans will hyperventilate when heat load becomes excessive [92]. At rest (during passive heating), pulmonary *V*E exceeds metabolic activity when T_c_ reaches 37.8–38.5 °C [93,94,95]. Above this temperature threshold, the ventilatory response is proportional to the degree of hyperthermia [93,94,95]. However, humans respond variably to heat stress [94], and the mode of heating influences their ventilatory response [96].

During exercise (active heating), the increased heat generated by contracting muscles and the resulting core hyperthermia act as additional ventilatory stimuli; an attenuation or amplification of exercise-induced elevation in T_c_ reducing [97,98] or increasing *V*E [38], respectively. There are, however, notable differences in hyperthermic hyperventilation between exercise and rest [94,95,96]. In some exercise-based studies, a temperature threshold for hyperthermia-induced hyperventilation has been reported, which might be intensity dependent (occurring at T_c_ ~37 °C during prolonged light-to-moderate [94,95] versus ~38 °C during incremental exercise to exhaustion [99,100]). Further, the sensitivity to increasing T_c_ appears variable between rest and exercise. At rest, elevations of T_c_ of 1 to 2 °C above normal levels lead to a rise in *V*E of ~3–5 L·min^−1^·°C^−1^ [93,96,101]. During constant-load, submaximal exercise in the heat, the rise in *V*E could, however, be larger; typical values ranging from ~4–10 L·min^−1^·°C^−1^ [38,94,96]. Finally, while an increase in breathing frequency primarily drives the rise in *V*E during exercise [94,102], sole or combined increases in breathing frequency and/or tidal volume contribute to hyperventilation in hyperthermic humans at rest [103,104].

Hyperthermia-induced hyperventilation has multiple consequences on the human body. Through excess elimination of CO_2_, hyperthermia-induced hyperventilation causes hypocapnia and respiratory alkalosis. At rest, hyperventilation associated with a ~1 °C gain in T_c_ causes a PaCO_2_ fall of ~10 mmHg and an elevation in blood pH to ~7.46 (with subsequent decrease in bicarbonate ions) [101]. During prolonged submaximal exercise in the heat (with ~2.5–3 °C gains in T_c_), PaCO_2_ commonly falls by ~5 mmHg [6,38]. When hyperthermic individuals get dehydrated during exercise, blood pH also increases (to ~7.43) and blood bicarbonate levels decrease [6]. During both active and passive heating, protection of blood homeostasis may therefore become secondary to the human’s thermoregulatory needs.

Reasons for hyperthermia-induced hyperventilation in humans are unclear. As in panting animals, an increase in *V*E that is primarily driven by tachypnoea could be a way (albeit small) for humans to increase evaporative heat loss through respiration [105]. Further, hyperventilation-derived heat exchange between the upper airways and the internal carotid artery may contribute to selective brain cooling [104,106]. However, that humans are capable of selective brain cooling remains hotly debated [107,108]. In fact, the reduction in PaCO_2_ associated with hyperthermia-induced hyperventilation may negatively affect the brain. PaCO_2_ has an important role in determining cerebral circulation [109]. In resting [59,96,110,111] and exercising [36,38] hyperthermic individuals, hypocapnia has been shown to contribute to cerebral hypoperfusion. As reduced brain blood flow during prolonged exercise with hyperthermia can affect convective heat removal from the brain to the body core [38], hyperthermia-induced hyperventilation could, via modulation of cerebral perfusion, challenge human cranial thermoregulation and homeostasis. However, the concomitant increase in extracranial blood flow [34,36,60,61] suggests that the net heat liberation from the human head might still be elevated in hyperthermic conditions where cerebral perfusion declines. Further work is required to elucidate the physiological significance of hyperthermic hyperventilation.

### 3.3. Breathing Mechanics and Control of Breathing during Exercise, with and without Environmental Stress

The increase in pulmonary ventilation during exercise is achieved by an increased mechanical work of the diaphragm and muscles of the rib cage and abdomen. Thus, exercise-hyperpnea/-hyperventilation generate extra metabolic and circulatory costs. Akin the other skeletal muscles, the respiratory muscles can fatigue. However, transient loss in force generating capacity of the inspiratory diaphragm [112] and expiratory abdominal muscles [113] only occurs when exercise is strenuous (>85% *V*O_2max_) and sustained (to the limit of tolerance). The impact of the transient loss in force generating capacity of the respiratory muscles on blood flow and O_2_ delivery to/consumption by the locomotor and respiratory muscles when *V*E is close to its peak has been the subject of extensive research and intense debate.

Fatigue of the respiratory muscles has been postulated to trigger a metaboreflex, leading to preferential ‘redistribution’ of blood flow to the diaphragm, and subsequent locomotor fatigue [114]. Growing evidence support (i) occurrence of a sympathetically-related vasoconstriction in limb locomotor muscles during exercise [11,64,115,116,117] and (ii) an influence of respiratory muscle work on locomotor muscle blood flow/fatigue (loading and unloading of the respiratory muscles having reciprocal effects on limb blood flow and endurance exercise time [118,119,120]). However, whether respiratory muscles have higher priority than locomotor muscles over blood flow is still an open question.

During maximal exercise, respiratory muscle blood flow might account for 14–16% of the systemic blood flow in highly trained athletes [121], with the oxygen cost of breathing estimated at 7–10% of whole-body *V*O_2_ in healthy untrained individuals, 13–16% in highly trained individuals [122,123], and upwards 20% in some endurance-trained women [123]. However, even in those highly trained individuals able to generate very high ventilatory flows during exercise (peak *V*E of ~165 L·min^−1^), the work of breathing only averages ~0.6 kJ·min^−1^ [123,124]. Based on Figure 1, the work of the respiratory muscles during maximal cycling would therefore represent only a very small fraction of whole-body aerobic energy turnover (which stands at ~98 kJ·min^−1^). Thus, the potential for respiratory muscles to ‘steal’ blood flow from locomotor muscles appears limited, if it ever happens. In contrast, for exercise intensities beyond 80% of maximum work-rate, a decline in blood flow and vascular conductance at the level of the respiratory intercostal muscle has been reported [124]. Further, through increases in O_2_ extraction from the circulation, torso and upper-body *V*O_2_ becomes elevated during high intensity exercise [64,117,125] (Figure 1). Although direct and continuous haemodynamic measures in the human diaphragm are still needed, evidence currently available does not provide strong support for a ‘prioritisation of blood flow’ toward the respiratory muscles during exercise.

How pulmonary *V*E is regulated during exercise is only partly understood and subject to much controversy [126,127,128]. Beyond humoral (blood-borne) factors, which can stimulate ventilation via activation of central (medullary neurons) and peripheral (carotid bodies) chemoreceptors, ventilatory changes during exercise seem heavily reliant on fast-acting neural feed-forward (i.e., an augmented central command [129]) and feedback from mechano- and metaboreceptors (particularly from locomotor muscle group III and IV afferents [130,131]). Depending on the phase of exercise (phase 1, 2 or 3 of hyperpnea), the intensity of exercise (leading to either hyperpnea or hyperventilation) and the environment (hypoxia, cold or heat exposure), the contribution—and possibly, interaction—of different signals in the drive to breathe is likely to vary.

The exact mechanisms underpinning hyperthermia-induced hyperventilation in humans are still disputed, with a likely differential contribution of body tissue temperatures (in particular, brain, muscles and skin temperatures) and of chemo- and metaboreceptors between rest and exercise. While at rest, an enhanced chemoreceptor ventilatory O_2_ drive partly contributes to the rise in *V*E [111], the involvement of carotid bodies during exercise seems negligible [132]. In exercising humans, at least three temperature-sensitive pathways could contribute to hyperventilation [133], i.e., (i) an increase in neural activity of the ventral respiratory group in the medulla [134]; (ii) an increase in efferent output from the cerebral cortex [135]; and (iii) an increase in neural discharge from muscle metaboreceptors (via group III and IV afferents) [136,137]. Further work needs to be done to establish the respective contribution of these different pathways (and possibly others) in the maintenance of blood homeostasis and cranial thermoregulation during hyperthermic hyperventilation.

## 4. Heart Function during Exercise and Environmental Stress

Exercise with and without environmental stress greatly increases tissue and organ systems requirements for O_2_ and nutrient delivery, and the removal of waste products and metabolically liberated heat. Up to moderate intensities, these demands are generally met by proportional increases in systemic blood flow (*Q*) and vascular conductance, with attenuated responses occurring during intense exercise or moderate exercise with extreme environmental temperature. The varied adjustments to exercise intensity and environmental stress provide insight into the control of cardiac function. This understanding is fundamental to determine whether the myocardium itself can maintain cellular homeostasis, and thus influences the development of fatigue during exercise with or without environmental stress.

### 4.1. Exercise Responses

From rest to highly intense exercise, the cardiovascular system adjusts the supply of O_2_ and nutrients to the energetic demand of the contracting skeletal, cardiac and respiratory muscles, but these responses are not uniform across all exercise modalities and intensities. During small muscle mass (i.e., single-leg knee extensor) and moderately intense dynamic exercise (i.e., two-legged cycling) increases in systemic blood flow and convective O_2_ transport are intensity dependent [11,13,64,138,139,140]—as evidenced by the linear rise in *Q* and systemic O_2_ delivery [13,64,125]. The constant perfusion-to-power and perfusion-to-*V*O_2_ relationships (e.g., 5–6 L·min^−1^·L O_2_) indicate that the cardiovascular system is operating well within its regulatory capacity during these exercising conditions (Figure 1) [11,13,125,141]. However, with whole-body exercise at higher exercise intensities, a mismatch between work rate and *Q* becomes evident [142,143]. As highlighted in Section 3, CaO_2_ is maintained or increased during strenuous exercise. The blunting in systemic O_2_ delivery during intense whole-body exercise is therefore related to alterations in systemic blood flow dynamics. Indeed, beyond ~40–50% of maximal aerobic work rates, increases in *Q* demonstrate a curvilinear response relative to increases in intensity [11,64,144,145], whereas the pressure gradient between the aorta and the right atrium keeps increasing compared to resting and low intensity exercise conditions [11,64,125,145,146]. The attenuation in *Q* is associated with an ever-increasing heart rate, a decline in stroke volume, and a plateau in systemic vascular conductance which all precede the attainment of *V*O_2max_ (Figure 1) [11,64,125,145,147,148,149]. Similar systemic cardiovascular strain is seen during maximal aerobic exercise at constant work rates [11,42,150]. It is therefore clear that maximal aerobic exercise in normal environmental conditions challenges the homeostatic control systems that govern the coupling of peripheral O_2_ demand and systemic O_2_ supply.

### 4.2. Impact of Adding Environmental Stress to Exercise

In warm environments, the additional thermoregulatory demands placed on the circulation by light or moderate exercise are generally well met. In this setting, elevations in skin blood flow to support convective heat transfer to the environment [151,152,153,154] are associated with >1 L·min^−1^ greater *Q* and moderately reduced arterial pressure compared to exercise in cool conditions [151,155,156,157,158]. Provided normal hydration status is kept, light and moderately intense exercise may continue for durations upwards of 2 h and is associated with stable *Q* and arterial pressure (Figure 2) [44,159,160,161,162,163]. However, the cardiovascular adjustments to exercise vary substantially, depending on the degree of environmental heat stress and/or exercise intensity. The development of whole-body hyperthermia in uncompensable hot environments is related to marked cardiovascular strain and impaired exercise capacity (Figure 3 and Figure 4). When exercise is performed above moderate intensities or in substantially elevated environmental temperatures, *Q* and stroke volume are suppressed in comparison to responses in cool conditions [12,66,164]. Furthermore, although variable among individuals and studies, reductions in *V*O_2max_ with whole-body hyperthermia generally occur in proportion to the magnitude of heat strain [41,67,150,151,165,166]. The restrictions in *Q* and O_2_ delivery that happen during incremental or maximal constant load exercise in normothermic conditions [11,150] display a remarkably similar, but accelerated response when exercise is combined with whole-body hyperthermia [35,150]. Hence, available evidence indicates that systemic blood flow, O_2_ supply and aerobic capacity are more rapidly impaired in exercise and environmental conditions causing severe whole-body hyperthermia.

Hydration status also modulates the effects that exercise intensity and thermal stress have on blood flow regulation. Studies manipulating total body water and body temperature at rest and during exercise using carefully controlled environmental conditions or water-perfused suits are indicative of this interaction. For instance, *Q* is maintained at rest and increases with small muscle mass exercise across graded levels of mild to moderate dehydration (~2–3.5% body mass loss) [167,168]. Furthermore, *Q* is relatively unaltered in individuals who perform moderate dynamic exercise in a cold environment in a dehydrated (up to ~4% body mass loss) versus a euhydrated state [157,169]. In contrast, a hallmark response to dehydration during prolonged moderate dynamic exercise in the heat is a progressive decline in *Q*, which is accompanied by reductions in stroke volume, arterial pressure and blood volume and concomitant increases in T_c_, heart rate and systemic vascular resistance (Figure 2) [6,43,44,162,164,170]. The marked impairments in *V*O_2max_ and endurance capacity also suggest that similar cardiovascular strain is apparent during maximal aerobic exercise [67,171,172]. For example, when maximal aerobic exercise is performed in a hyperthermic state, both with and without dehydration, *V*O_2max_ and endurance capacity are reduced to a similar extent; ~16% and ~52% relative to control conditions, respectively (Figure 3) [67]. Taken together, the experimental evidence highlights the potential for whole-body hyperthermia with and without dehydration to speed up the reductions in systemic O_2_ delivery and impair aerobic capacity and endurance performance (Figure 4). A key question however, is what mechanisms underpin the impaired systemic blood flow?

### 4.3. Cardiac and Peripheral Mechanisms Regulating the Systemic Circulation

The contributions of the heart and the periphery to the regulation of systemic blood flow during exercise and environmental stress is an area of enduring debate within integrative physiology. The prevailing theory views the heart as a ‘pressure-propulsion pump’. Under this paradigm, the heart is the pump that generates the total driving force (pressure) for the blood’s propulsion through the peripheral vessels and its distribution to the brain, the respiratory, cardiac, postural and locomotor muscles, the skin, and the splanchnic and renal vascular beds [3,173,174,175,176,177,178,179]. In this model, cardiovascular function is predominantly determined by intrinsic factors (such as myocardial contractility) and by extrinsic factors that alter the preload and afterload of the heart (i.e., venous return, mean arterial pressure and peripheral vascular resistance). Venous flow (venous return to the pump) is also driven by a pressure gradient between the venous vasculature of peripheral tissues and the right atrium, and is facilitated by the action of the skeletal muscle pump [180,181,182,183,184]. Experimental findings will be interpreted in the context of this paradigm to answer whether the function of the heart itself is compromised during exercise with and without environmental stress. Alternative views will be presented in the integrative section of the review.

Direct and indirect evidence of the intrinsic chronotropic, inotropic and lusitropic responses to exercise and heat stress sheds light into myocardial function during conditions of increased energetic demands. It is well established that exercise heart rate, contractility (via β-adrenergic stimulation) and stroke work all increase the O_2_ demand of the myocardium [185,186,187,188,189,190,191,192]. However, a crucial question is whether the heart reaches its functional capacity during maximal aerobic exercise. Limited direct evidence from healthy humans exercising in cool ambient conditions reveal that coronary artery blood flow increases three- to five-fold above resting levels in proportion to elevations in heart rate [190,193]. A substantial coronary flow reserve may also be present at exhaustion, as indicated by elevated flows during exercise in hypoxia at lower absolute workloads compared to normoxia [194,195]. Concurrently, coronary venous O_2_ saturation decreases from ~33% to ~24% with intense exercise, indicating that coronary venous O_2_ content declines and arteriovenous oxygen differences increase, despite coronary venous O_2_ tension being largely unaltered during exercise > 85% of maximal heart rate [186,194]. These limited observations imply myocardial *V*O_2_ may be sufficiently increased during maximal aerobic exercise. In this light, the rate pressure product—a widely accepted index of myocardial workload (i.e., product of heart rate and systolic blood pressure)—also rises linearly during incremental dynamic exercise to volitional exhaustion [11,148,149,192]. Using right atrial pacing throughout incremental exercise to increase heart rate by ~20 beats·min^−1^ above normal values, Munch et al. [125] demonstrated that maximal *Q* is unaltered due to reciprocal reductions in stroke volume. Arterial, central venous and pulmonary pressures and systemic vascular conductance were also remarkably similar to control exercise [125]. The findings strongly indicate that peak myocardial *V*O_2_ can be increased during maximal aerobic exercise without negatively impacting the overall cardiac performance. Consequently, the human heart appears to be working below its capacity during maximal aerobic exercise, and thus limitations in myocardial metabolism and work capacity do not seem to explain the blunted or reduced systemic blood flow and vascular conductance.

Recent advancements in non-invasive imaging and speckle-tracking echocardiography also shed light into the intrinsic and extrinsic factors that may modulate the myocardial contractile function during exercise and heat stress with and without dehydration. Firstly, studies assessing left ventricular mechanics establish the influence of intrinsic cardiac factors. Following exercise-induced dehydration (with a ~3.5% loss of body mass), resting tissue Doppler derived measures of systolic and diastolic function are maintained [196], while the systolic mechanical twisting and rotational velocity of the left ventricle—made possible by the helical orientation of cardiac myofibres—is enhanced compared to euhydrated responses in the face of a reduced stroke volume [167]. Despite this, resting *Q* is unaltered or elevated [167,196], indicating overall cardiac performance is not impaired. When exercise is performed in a dehydrated and hyperthermic state, left ventricular systolic rotation, rotational velocities and twist, as well as diastolic untwisting rate are maintained or enhanced [44,167,197]. Furthermore, end-systolic volume has been shown to be maintained [197] or to decrease slightly [44,167] in these conditions, suggesting contractility is preserved or may even be enhanced during exercise, dehydration and hyperthermia.

Secondly, the impact of afterload on cardiovascular function during exercise and heat stress can be explored from the relationships between left ventricular volumes and arterial pressure/systemic vascular resistance. Expression of these relationships as effective arterial elastance—an index of the net arterial load imposed on the left ventricle [198]—and end-systolic elastance—an integrated measure of left ventricular performance [198,199]—provides additional novel insight into the potential peripheral factors that may underpin cardiovascular function. With incremental exercise in normothermic conditions, systolic twisting of the left ventricle increases [145] and end-systolic volume is maintained as exercise intensity increases from moderate to heavy/severe intensities despite elevated systolic pressure [145,148,149,192]. Afterload also appears to have minimal influence on the decline in *Q* with progressive exercise-induced dehydration in the heat, as arterial blood pressure is similar or slightly reduced compared to euhydrated responses [6,43,44,162,170,197,200]. However, parallel increases in effective arterial elastance have been observed during prolonged exercise [44,197], suggesting a possible contribution of increased afterload on lowering stroke volume. Notwithstanding, the simultaneous observations that end-systolic elastance is maintained and end-systolic volume may be slightly reduced [44,197,200] indicate that the impact of increased afterload on stroke volume is negligible. Together, these observations suggest afterload does not impair the heart’s ability to eject blood into the systemic circulation during intense exercise, whole-body hyperthermia and moderate dehydration.

Characterisation of left ventricular volumes during exercise and heat stress addresses the impact of preload on cardiac performance. It has been postulated that a reduction in preload when exercise is performed in the heat is related to the increasing thermoregulatory demands for skin blood flow [12,138,151,201]. Under this paradigm, decrements in stroke volume occur following redistribution of centrally circulating blood to compliant cutaneous vascular beds. This acts to lower ventricular filling pressure and—in the face of a reduced stroke volume—heart rate is elevated in an attempt to maintain *Q* [12,138,151,201,202]. In this construct, reductions in systemic blood flow observed during exercise-induced dehydration may simply be a consequence of a reduced *Q*. Support for this notion stemmed from observations that whole-body cooling reverses stroke volume and central venous pressure responses seen with whole-body hyperthermia and exercise [12]. However, not only does aggressive cooling influence cutaneous blood flow, but it also lowers skin and core temperatures. Furthermore, skin perfusion reaches maximal rates during dynamic exercise in the heat when T_c_ approaches ~38 °C [152,163,203], and yet stroke volume continues to decline as significant hyperthermia develops (i.e., T_c_ substantially above 38 °C) [66,204,205,206]. Hence, skin temperature appears to play a relatively minor role in comparison to T_c_ on the stroke volume responses to exercise in hot conditions [207].

An additional possibility that relates to the intrinsic activity of the heart is the heart rate itself. Increasing exercise intensity or elevations in T_c_ and sympathoadrenal activity elevate heart rate during exercise [157,162,170,208], resulting in a shorter duration of diastolic filling and concomitant fall in end-diastolic volume [145,167,204,206,209]. These responses may be exacerbated further by dehydration and the associated concomitant hyperthermia and reduced circulating blood volume [44,197,200]. Experiments manipulating heart rate and blood volume during exercise in the heat with and without dehydration support the notion of a dual role in impairments in diastolic filling. By preventing the typical rise in heart rate during exercise in the heat via β1-adrenergic blockade, stroke volume is maintained rather than reduced [204,206,209] (presumably via increased diastolic filling time). Furthermore, intravenous infusion of dextran at a rate matching expected sweat losses during exercise-induced dehydration results in a maintained blood volume, while the typical progressive elevation in heart rate is largely unaltered [203]. Interestingly, however, this approach reduces the decline in stroke volume by ~50% to that observed during exercise without any form of fluid replacement [203], and completely restores stroke volume and heart rate to euhydrated levels when body temperature is maintained by exercise in the cold, despite persistent intracellular and interstitial dehydration [210]. Taken as a whole, these findings suggest that a combination of tachycardia and reduced blood volume reduce stroke volume with exercise, heat stress and dehydration.

The collective findings outlined above suggest the functioning of the myocardium itself is well maintained in the face of the profound physiological strain induced by intense exhaustive exercise in normal and hot environmental conditions. However, it appears that several peripheral factors, extrinsic to the myocardium, are associated with attenuations in systemic blood flow and vascular conductance. The control—and potential consequences of impaired—matching of O_2_ and nutrient delivery with removal of metabolic waste and heat in the locomotor muscles is discussed in detail in the next section.

## 5. Skeletal Muscle Function during Exercise and Environmental Stress

Skeletal muscle contraction produces the force and power necessary for locomotion. The underpinning formation and dissociation of cross-bridges between the myosin and actin myofilaments require chemical energy generated through adenosine triphosphate (ATP) hydrolysis. Because of its limited cellular storage, ATP is continuously resynthesised via phosphocreatine hydrolysis, glycolysis and oxidative phosphorylation, but their contributions primarily depend on the interplay of exercise intensity and duration [211,212,213,214]. During exercise longer than 1–2 min, however, the majority of ATP is generated in the muscle mitochondria via oxidative phosphorylation, using reducing energy equivalents from carbohydrate and fat metabolism [212,215,216]. Close matching of O_2_ supply and demand is therefore crucial to sustain muscle metabolism and contractile function during the majority of exercise modalities. The following sections will address whether O_2_ supply to contracting skeletal muscle is tightly regulated across a range of exercise and environmental conditions.

### 5.1. Exercise Responses

It is well established that blood flow to working muscles in humans increases with elevations in exercise intensity at an average rate of 75–80 mL·min^−1^·W^−1^ [5,13,217,218]. This local response is met by a similar rise in systemic blood flow, but a modest increase in the arteriovenous pressure gradient [11,64,65,117,125]. The magnitude of increase in skeletal muscle perfusion can reach 100- to 160-fold in untrained and trained humans performing small muscle mass exercise (i.e., an increase from ∼2.5 at rest to ∼250 and ∼400 mL·min^−1^·100 g^−1^ at maximal single leg dynamic knee-extensor exercise in untrained and trained individuals, respectively) [13,217,219]. These estimates of maximal perfusion in human skeletal muscle agree with values in animal studies [220,221,222] and demonstrate the enormous capacity of the skeletal muscle vasculature to dilate and increase blood flow when aerobic metabolic demand increases. The delivery of O_2_ and nutrients to working skeletal muscle and the clearance of metabolic waste products and heat are a function of blood flow, the blood concentration of gases, substrates and metabolites, and the temperature of the blood. The concentration of O_2_ and nutrients in the arterial blood is typically maintained or changes slightly during short duration exercise in normal conditions. Blood flow is therefore the primary determinant of O_2_ and fuel delivery to active skeletal muscle during dynamic exercise [4,219,222,223,224]. Similarly, venous blood flowing from the locomotor limbs towards the heart and the lungs transports CO_2_ and heat to be released to the environment during the processes of pulmonary gas exchange and cutaneous heat liberation. While a small amount is released via the respiratory system, most of the heat liberation to the environment occurs when blood circulates through the cutaneous circulation, in particular in the locomotor limbs [66,225,226,227].

Mechanisms sensing the rate of muscle O_2_ utilisation and heat production are thought to largely control the magnitude of increase in blood flow to active skeletal muscle, also known as exercise hyperaemia [4,219,222,225]. The progressive increases in blood and tissue temperature, leg blood flow, leg O_2_ delivery, leg O_2_ extraction and leg *V*O_2_ in healthy young men during incremental single leg dynamic knee-extension exercise and submaximal cycling supports this idea [11,13,64,65,117,225,228]. These circulatory adjustments across the active leg tissues—chiefly, the exercising quadriceps muscle in the knee-extensor exercise model [13,228,229]—are for the most part accompanied by proportional increases in *Q*, systemic O_2_ delivery and *V*O_2_. Yet an exponential rise in these variables occurs when engaging the upper body muscles before reaching exhaustion during this small muscle mass exercise model [64,65,125] (Figure 1). Accordingly, a primary aim of the peripheral and central mechanisms involved in the regulation of the circulation during dynamic steady-state exercise is to maintain a close match between O_2_ supply and demand in active muscle [4,219,222,223,224,230]. However, close matching of O_2_ supply and utilisation is not a universal physiological phenomenon in active skeletal muscle across all exercise modes and intensities.

Evidence in human models indicates that blood flow and O_2_ delivery to active locomotor skeletal muscles is restrained when engaging a large muscle mass during intense whole-body exercise, e.g., cycling, skiing, rowing, or running [8,11,35,64,116,117,125,231,232,233]. In support of this, leg blood flow and O_2_ delivery have been shown to be the same up to 40% of the maximal aerobic work rate (W_max_). Moreover, perfusion and O_2_ supply are intimately related to leg *V*O_2_, irrespective of whether power output is generated only by the quadriceps muscles during single leg knee-extensor exercise or spread among the different leg muscles during two leg cycling [64,125] (Figure 1). These observations are strong evidence of the intimate coupling between O_2_ delivery and metabolic demand. Nonetheless, during cycling exercise above a moderate intensity, the relationship between locomotor limb tissue perfusion/O_2_ supply and O_2_ utilisation is curvilinear [64,125]. This curved relationship in the moderate to maximal intensity domain is indicative of a mismatch between O_2_ supply and metabolic demand for O_2_ [64,125] (i.e., above 50–80% W_max_; Figure 1 and Figure 5). As a consequence, muscle perfusion per unit of power is lower during maximal whole-body exercise than during maximal small-muscle exercise (i.e., single leg knee-extensor exercise) [64,125,234]. Importantly, the substantial rise over time in active muscle iEMG activity during exhaustive exercise [77,80,235,236,237,238] suggests that the blunting in active muscle perfusion (i.e., blood flow per unit of active muscle) is actually greater than that revealed from the limb blood flow to power relationship, since muscle recruitment increases out of proportion to power output. Regardless of the true magnitude of the phenomenon, it is clear that the observed restrictions in contracting muscle blood flow impose a circulatory limitation to aerobic capacity and endurance performance.

In contrast to the well-characterised peripheral and systemic haemodynamic responses during short duration exercise, and due to the narrow applicability of the available invasive and non-invasive methodology, relatively limited data exist on the circulatory response of the working muscles to prolonged whole-body exercise. Using the constant-infusion thermodilution technique, studies reveal stable exercising limb blood flow during prolonged cycling when hydration status is maintained [6,155,156,239,240,241,242,243] (Figure 2). Direct measurements and estimates of blood flow in non-exercising tissues and organs (upper body) blood flow during prolonged exercise are also suggestive of stable brain, upper limb, splanchnic and renal circulations [6,43,44,239] (Figure 2), as well as body temperature, metabolic, fluid and ion balance and acid-base profiles [66,161,162,170,239,244]. Hence, matching of O_2_ supply and demand, and maintenance of thermal and acid-base balances can happen at the local and systemic levels during prolonged exercise even in warm conditions as long as body fluid status is preserved.

### 5.2. Impact of Adding Thermal Stress to Exercise

As discussed above in relation to the responses to small and large muscle mass exercise in normal conditions, exercise type—or more specifically, the overall physiological demand—acts as an important modulator of the skeletal muscle adjustments to separate and combined exercise and heat stress. Studies utilising small muscle-mass exercise (e.g., single-leg knee-extensor exercise) demonstrate an elevation in limb blood flow during exercise with heat stress, suggesting an additive effect of metabolic and thermoregulatory blood flow requirements when whole-body physiological demand is very low (power output 10 to 50 W for a trained individual) [158,245]. Even at these low absolute exercise intensities, however, the elevation in limb blood low is lower than would be expected given the magnitude of hyperaemia observed when heat stress and exercise are studied in isolation. As an example, while severe passive hyperthermia results in elevations in leg blood flow of ~1 L·min^−1^ [158,245,246,247,248,249], the superimposition of the same thermal stimulus onto single-leg knee-extensor exercise in the same individuals results in increases of only ~0.5 L·min^−1^—an attenuation of 50% of the response seen at rest [158,245,250]. When a larger muscle mass is engaged (e.g., cycling), the hyperthermia-induced augmentation in exercising limb blood flow disappears altogether [35,155,156,240,241] (Figure 5). This suggests that the much greater metabolic and thermal effects of whole-body exercise on exercising leg blood flow (~16–19 L·min^−1^ at peak exercise) [35] cancel out the independent influence of hyperthermia observed during passive heating or small muscle mass exercise.

The skeletal muscle circulatory responses to combined dehydration and hyperthermic stress are also different at rest and during isolated-limb exercise compared to strenuous whole-body exercise. At rest, limb blood flow and limb vascular conductance (limb blood flow/limb perfusion pressure) are enhanced with progressive dehydration and hyperthermia; a response maintained during small muscle mass exercise, despite substantial body mass losses (~3.5%) and reductions in limb perfusion pressure [168]. Conversely, during prolonged whole-body exercise, there is a substantial ~2 L·min^−1^ reduction in locomotor limb blood flow compared to control exercise and a ~1.3 L·min^−1^ combined reduction in skin, brain, and visceral organ blood flow (Figure 2) [6,43,44,162,203]. Similarly, the combination of heat stress and whole-body exercise at near maximal intensities (e.g., 80–100% *V*O_2max_) poses a major challenge to cardiovascular control. Severe hyperthermia results in a quicker onset of fatigue, due to a more rapid reduction in blood flow to the working skeletal muscles of the locomotor limbs compared to exercise at temperate ambient temperatures [150] (Figure 3 and Figure 4). This muscle hypoperfusion leads to reduced convective O_2_ delivery and blunted aerobic metabolism, because maximal functional O_2_ extraction of the exercising limb tissues (~90–95%) is achieved in the early stages of severe-intensity exercise [150]. Ultimately, the ensuing mismatch between O_2_ supply and demand compromises skeletal muscle metabolism, contractile function, power output and endurance capacity.

### 5.3. Mechanisms of Blood Flow Control—Impact of Exercise, Hyperthermia and Dehydration

According to the hydraulic equivalent of Ohm’s law, skeletal muscle perfusion is determined by vascular conductance (the inverse of resistance) and perfusion pressure gradient, i.e., arteriovenous pressure difference. During dynamic incremental exercise, the increases in active limb muscle blood flow are primarily the result of increases in vascular conductance. The latter is indicative of vasodilatation of the vascular beds perfusing the active muscle, as the perfusion pressure gradient does not change during mild intensity exercise, or increases slightly, in comparison to flow during intense exercise. In exercising limbs, the diameter of the large conduit vessels, such as the femoral and brachial arteries, does not appear to change, or changes little, in comparison to the increases in arterial blood velocity [158,168,218,245,251,252,253,254]. Consequently, the enhanced vasodilatation, indicated by the global increases in limb vascular conductance during exercise, is attributable largely to increases in the diameter of small arteries and arterioles perfusing active muscle fibres and rheological changes underpinning a decrease in blood viscosity with, among other factors, increases in local temperature [230,255,256,257,258].

Several local and central control mechanisms have been proposed to regulate active muscle blood flow. These include metabolic, myogenic, mechanical, humoral and neural mechanisms, which involve the interaction of multiple intravascular, interstitial and intracellular signaling pathways [3,4,222,223,259,260]. The close coupling between O_2_ supply and utilisation across a wide range of conditions supports a role of oxygen sensing mechanisms in the so-called metabolic vasodilation [255,261]. In particular, red blood cells (RBCs, which constitute ~84% of all human cells in the body by number) [262] contribute to the matching in O_2_ supply and demand in skeletal muscle during dynamic exercise by three O_2_-dependent mechanisms: (i) release of the vasoactive substances ATP, (ii) S-nitrosohaemoglobin (SNO-Hb)-dependent vasodilation, and (iii) reduction of nitrite to vasoactive NO [19,255,261,263,264,265]. The regulatory role of RBCs in humans is strongly supported by the observations that, (i) exercising skeletal muscle blood flow responds primarily to changes in the amount of O_2_ bound to the erythrocyte haemoglobin molecules, rather than the amount of O_2_ in plasma [251,266,267,268,269,270,271], and (ii) exercising muscle blood flow during submaximal single leg knee-extensor exercise can almost double (from 260 to 460 mL·min^−1^·100 g^−1^) with marked acute alterations in blood O_2_ content (i.e., with combined polycythaemia and hyperoxia vs. combined anaemia, plasma volume expansion and hypoxia), such that O_2_ delivery and *V*O_2_ are kept constant [272]. Exercise is associated with decreases in intracellular O_2_ markers [141]. It therefore seems plausible that exercise hyperaemia is also influenced by concomitant activation of intracellular O_2_ sensing mechanisms [4,273]. Intracellular O_2_ markers such as *P*O_2_ and myoglobin O_2_ saturation cannot, however, explain the further elevation in exercising limb muscle blood flow occurring with carbon monoxide-induced arterial hypoxaemia. This is because quadriceps muscle *P*O_2_ and myoglobin O_2_ saturation remain at similar levels compared to normoxia, along with normal venous and arterial *P*O_2_ levels [274]. Thus, exercising limb blood flow responses to independent alterations in blood oxyhaemoglobin and plasma free O_2_ are unrelated to arterial, venous or muscle *P*O_2_ [266,267,268,274]. The main vascular O_2_ sensor locus for the control of blood flow therefore appears to lie in the erythrocyte itself, rather than in the *P*O_2_-sensitive areas of the vascular endothelium, vascular smooth muscle or skeletal muscle. These data collectively support that O_2_ delivery, rather than tissue blood flow per se, is a key regulated variable involving in part O_2_ sensors in the erythrocytes and the concomitant release of vasoactive substances into the vascular lumen. These vasoactive signals lead to relaxation of vascular smooth muscle and vasodilatation by binding to specific receptors in the vascular endothelium and triggering downstream signal transduction pathways [222,255,259,260,273].

Regulation of O_2_ supply during exercise and environmental stress not only involves locally released vasodilator signals, but also vasoconstrictor signals tied to the increase in muscle sympathetic nerve activity and circulating vasoconstrictor substances (e.g., noradrenaline) [4,259,273,275]. This construct can be used to explain the differential limb hemodynamic responses to dehydration and/or hyperthermia under resting and different exercising conditions (i.e., an increase at rest and small muscle mass exercise vs. a decline during whole-body exercise). In conditions of small muscle mass exercise with dehydration and hyperthermia, it is possible that thermal, fluid and oxygen sensing mechanisms activated by (i) increases in local tissue temperature similar to that observed during local and whole-body passive heating [158,245,246,247,276,277] (ii) changes in red blood cell deformability and cell volume [278,279] and (iii) elevations in arterial oxygen content concomitant to the dehydration-mediated haemoconcentration [19,168] lead to augmented vasodilator activity in the face of a very low systemic sympathetic activity (e.g., circulating noradrenaline 1.2 to 1.9 nmol·L^−1^). This contrasts with the responses to whole-body prolonged exercise where a similar fall in mean arterial pressure (~8%) is accompanied by a much larger increase in circulating catecholamines (~18 nmol·L^−1^). Paradoxically, however, limb vascular conductance is essentially unchanged or slightly enhanced [6,168]. This observation is similar to those seen during constant work rate, maximal aerobic exercise with superimposed body hyperthermia and high levels of circulating catecholamines [150] (see Figure 4). However, this finding disagrees with those seen during incremental cycling to exhaustion where exercising leg vascular conductance is consistently blunted at near to maximal aerobic exercise intensities and muscle sympathetic nerve activity and circulating catecholamines are increasing exponentially to values many-fold higher than at rest or low intensity exercise [11,35,64,115,116]. These findings illustrate the regulatory complexity underlying the locomotor muscle blood flow responses to exercise and environmental stress, which differ depending upon the exercise type and intensity; factors determining the magnitude of stress imposed on the body. In conditions requiring a high fraction of the functional capacity, the restricted contracting skeletal muscle blood flow might not only be mediated by augmented vasoconstrictor activity, but also by suppressed vasodilator function at the level of the microcirculation; a possibility that warrants investigation.

## 6. Integrative View of Physiological Systems and Homeostasis during Exercise and Environmental Stress

From the above discussions regarding the responses of individual organ systems to exercise with and without heat stress, it is evident that the ability to perform exercise in different environments is greatly dependent on the coordinated functions of the human brain, the lungs, the heart, and the muscles. The functional responses of these organs are underpinned by a myriad of mechanisms encompassing (but not limited to) locally released regulatory substances and signals discharged by the endocrine, autonomic and central nervous systems. Discussion of the roles and specific contributions of the neuroendocrine systems in this context is beyond the scope of this review. The reader is instead directed to other excellent reviews of these topics [3,4,138,280,281,282]. In this section, we present two scenarios to explain the integrative physiological responses to exercise and environmental stress, and discuss how these vital organ systems interact to maintain cellular homeostasis and function in most exercise conditions. The first scenario exemplifies the exercise and environmental stress conditions where physiological steady state is achieved (Figure 6). In this scenario, increases in the rate of locomotion and the degree of thermal stress induce multiple functional and homeostatic adjustments, which primarily involve: elevations in O_2_ and substrate demand utilisation, and increases in CO_2_ and heat production mostly in the working skeletal muscles. The associated rise in muscle blood flow ensures appropriate O_2_ and substrate supply and CO_2_ and heat removal to the surrounding environment via the circulation. These peripheral responses are linked to the appropriate increase in central neuromotor drive, pulmonary ventilation, systemic blood flow and arterial pressure, and are supported by elevations in central and autonomic neural activity, as well as global respiratory, vascular, and cardiac functions. Cellular homeostasis is preserved across all organ systems, as the physiological demands of exercise and thermal stress are substantially below the body’s functional capacity; consequently, a steady state is achieved. Contrastingly, the second scenario (Figure 7) illustrates the physiological strain occurring during near or maximal aerobic exercise with and without environmental heat stress. The herein reviewed evidence suggests that compensatory adjustments seemingly preserve cellular homeostasis and function in the brain, the lungs, and the heart (as evidenced by the maintenance or increase in the corresponding regional *V*O_2_, central drive, pulmonary *V*E and heart rate). However, a mismatch between energy demand and supply in contracting skeletal muscle leads to compromised local metabolism, suppressed muscle contractile function, diminished force production and, ultimately, an inability to maintain speed of locomotion or power output (i.e., the hallmark of fatigue). In this view, reductions in skeletal muscle blood flow underlying the compromised O_2_ delivery and *V*O_2_ play a central role in the chain of events leading to fatigue.

The Fick principle is used in this review to establish whether exercise and environmental stress are met by either physiological steady state or strain. This integrative approach provides insights into the differing organ systems requirements of the body’s functional capacity and thus, the varied influence of exercise and environmental stress in each system. The comparison of the adjustments to prolonged and maximal aerobic exercise with and without significant dehydration and hyperthermia helps identify common physiological perturbations but different homeostatic challenges among organ systems. Disparities in their functional capacities and regulatory mechanisms governing local O_2_ supply, tissue O_2_ extraction and O_2_ utilisation might explain the differential organ system responses. The physiological strain associated with the impairment in endurance capacity during prolonged and maximal aerobic exercise is typified by: hyperventilation; internal body hyperthermia; blunted *Q*, restricted blood flow in the active skeletal muscles, respiratory muscles, skin, brain and visceral organs; enhanced total peripheral resistance; and a small reduction in mean arterial pressure or perfusion pressure [6,9,11,44,64,66,82,83,117,124,125,150,197]. During submaximal exercise, the reductions in peripheral blood flow and O_2_ supply are compensated for by increases in tissue O_2_ extraction, such that whole-body and regional *V*O_2_ are preserved [6,66]. Conversely, severe physiological strain during maximal aerobic exercise impairs *V*O_2max_ in proportional to reductions in locomotor muscle *V*O_2_ [11,64,117,125,150]. The attenuation in the rate of O_2_ delivery is temporally associated with the attainment of the functional limit for O_2_ extraction across the locomotor limbs (~90–95% of O_2_ extraction reserve used), ensuing in a blunting in active skeletal muscle *V*O_2_ and *V*O_2max_ [11,64,125,150,283]. In contrast, the brain, the lungs and the heart do not appear to attain the limit of their functional O_2_ reserve at exhaustion, and thus organ *V*O_2_ is preserved or increased [36,42,43,117].

A common phenomenon described herein is the blunting of peripheral and systemic blood flow, leading to attenuated O_2_ delivery to the working muscles, the brain, and other peripheral territories during near-to-maximal aerobic exercise intensities in varied environmental conditions and during prolonged intense exercise with significant dehydration and hyperthermic stress. These temporally linked restrictions in skeletal muscle blood flow and *Q* give rise to the alternative possibility that regulatory events in the peripheral microcirculation play a crucial role in the output of the heart [44,125,284,285,286]. Several empirical observations support this hypothesis: (i) artificial pacing of heart rate at rest and during exercise up to maximal aerobic capacity leaves *Q* and exercising limb blood flow unaltered, suggesting that tachycardia independent of peripheral circulatory events is inconsequential to activity of the heart [125,204,209,287,288,289]; (ii) incremental exercise in a man with an implanted cardiac pacemaker was associated with substantial elevations in end-diastolic volume, stroke volume, ejection fraction, pulmonary wedge pressure and *Q*, despite constant heart rate at 100 beats·min^−1^, whilst systemic blood flow rose in relation to *V*O_2_ up to 2 L·min^−1^ [3] (p. 181); (iii) increases in conduit artery blood flow during single leg knee-extensor exercise and passive segmental leg heating are accompanied by selective increases in downstream blood flow [246,290]. This is similar to what happens during intense whole-body exercise across the head, with brain blood flow declining when extracranial perfusion increases [35,36]; (iv) blood flow in non-exercising organs and tissues (including in the arm and brain) are relatively unchanged during incremental and constant power exercise, when large increases in *Q* occur (Figure 1 and Figure 5). This evidence ((iii) and (iv)) suggests that the heart does not determine local blood flow distribution; (v) in ex vivo pressurised preparations of small muscle resistance arteries, pharmacologically-induced alterations in vascular tone evokes substantial changes in blood flow in the absence of a heart [261,291,292]; (vi) infusion of vasodilator or constrictor substances into the femoral artery at rest and during submaximal and maximal aerobic exercise leads to proportional augmentation or blunting in leg blood flow and *Q*, respectively, but no changes in brain or contralateral limb blood flow [65,116,287,293,294,295]; (vii) infusion of ATP into the femoral vein (at a rate that causes substantial leg and systemic hyperaemia when infused in the femoral artery), does not change either leg blood flow or *Q* [65]; (viii) limb blood flow per unit of power is remarkedly similar during different exercise modalities despite very different systemic blood flows [64,125]; (ix) likewise, in the face of large differences in *Q*, limb blood flow is highly comparable during passive whole-body heat stress and isolated whole-limb and limb segment heating [158,246,247,277]; (x) the reductions in peripheral blood flow associated with exercise-induced dehydration parallel the diminished venous flow to the heart, suggesting that the lowering of *Q* occurs only when there is an interaction among reduced peripheral blood flow, dehydration-induced hypovolemia and tachycardia [6,44,210]; and (xi) exercising limb blood flow and *Q* per unit of power are restricted during exercise intensity domains above *V*O_2max_, yet the overall circulatory response is closely tied to contracting muscle aerobic metabolism [259,296]. These observations underscore the need for new integrative models that better explain the regulation of the heart and circulation across the whole range of exercise intensities and environmental stresses that humans can undertake/are exposed to.

## 7. Organ Systems Interactions—Future Research Directions

An alternative to the cardio-centric model considered in this review is that the regulation of the peripheral circulation, rather than the activity of the heart per se, determines the increase in *Q* and exercising muscle blood flow during exercise [44,125,284,285,286]. From a biophysical viewpoint, the heart can be viewed in this unconventional model as an organ of impedance whose mechanical function (comparable to a hydraulic ram) maintains pulmonary and arterial pressure through the cyclic interruption of flow [284,297,298,299]. Another radical idea supported by the profound increases in leg blood flow and *Q* during pharmacologically induced vasodilatation (7–8 L·min^−1^) in absence of the muscle pump, changes in metabolism or alterations in perfusion pressure in resting upright seated humans [65], is that the blood possesses autonomous movement [299,300,301]. Under exercise and environmental stress, blood movement through the circulation can be sustained by regulatory pathways that adjust local vascular conductance (and thus, blood flow) according, chiefly, to: metabolic and thermoregulatory demands, and acid-base homeostatic needs of organs and tissues [246,255,302].

In terms of brain–respiratory interactions, recent developments in neuroscience highlight the potential for astrocytes of the brainstem chemoreceptor areas to act as key cellular sensors in the regulation of breathing (particularly in conditions of increased metabolic demand, such as during exercise). Through detection of changes in brain tissue homeostatic parameters (in particular, pH/CO_2_ [303]), astrocytes could modulate the activity of medullary neurons essential for respiratory rhythmogenesis. In conscious rats, blockade of astrocyte vesicular release from the pre-Bötzinger complex (i.e., the neural network thought to play a central role for respiratory rhythmic generation [304]) has been shown to: alter breathing at rest; impair responses to hypoxia and hypercapnia; and dramatically reduce exercise capacity [305]. We may therefore posit that, in the exercising human, astroglial signalling strengthens the interactions between the brain and the lungs, and that it assists (via adjustment in the ventilatory response) in the maintenance of homeostasis.

The potential contribution of brain function to locomotor muscle fatigue is a subject with polarised views in the literature. For instance, the ‘central governor’ model of fatigue postulates that an anticipatory feedforward mechanism terminates strenuous exercise by depressing central motor drive before skeletal muscle blood flow and *Q* are compromised [306,307,308]. This idea, however, is at odds with empirical studies revealing that skeletal muscle blood flow and *Q* decline before muscle *V*O_2_ and *V*O_2max_ are compromised, and before exhaustion occurs [11,64,117,125,150]. In this line, the rise in active muscle iEMG activity during exhaustive exercise [77,80,235,236,237,238] hints that central drive is increasing rather than decreasing prior to exhaustion. Whether the brain contributes to fatigue processes via other mechanisms deserves further research.

## 8. Conclusions

This review highlights the integrated functional homeostatic adjustments of multiple organs and tissues to elevations of O_2_ demand and consumption, and CO_2_ and metabolic heat production in response to exercise and environmental stress. The coordinated activity of the brain, lungs, the heart and skeletal muscles is capable of providing adequate increases in central neuromotor drive, pulmonary ventilation, systemic blood flow and arterial pressure under conditions of low physiological load. These conditions include small muscle mass exercise and prolonged submaximal dynamic exercise in warm compensable environments with adequate fluid intake (Figure 6). However, the physiological strain imposed by severely intense dynamic exercise (e.g., running, cycling, rowing, swimming) is associated with insufficient blood flow and O_2_ delivery to the exercising limbs, resulting in a mismatch between energy demand and supply in the contracting muscles. This manifests as compromised skeletal muscle metabolism, suppressed muscle contractile function, impaired force production, and, ultimately, an inability to maintain speed of locomotion or power output (i.e., fatigue) (Figure 7). This impaired homeostatic regulation of skeletal muscle metabolism to intense exercise is accelerated by the superimposition of whole-body hyperthermia and dehydration, while function and *V*O_2_ of the brain, lungs and heart are maintained. The impact of exercise and environmental stress upon whole-body functions and homeostasis therefore depends on the functional needs, and differs across organ systems. In sum, Claude Bernard’s concept of stability of the internal milieu—the underlying principle of homeostasis—remains highly relevant to the understanding of physiological adjustments, and functional homeostatic differences among organ systems during exercise and environmental stress in humans.

## Figures and Tables

**Figure 1 cells-11-00383-f001:**
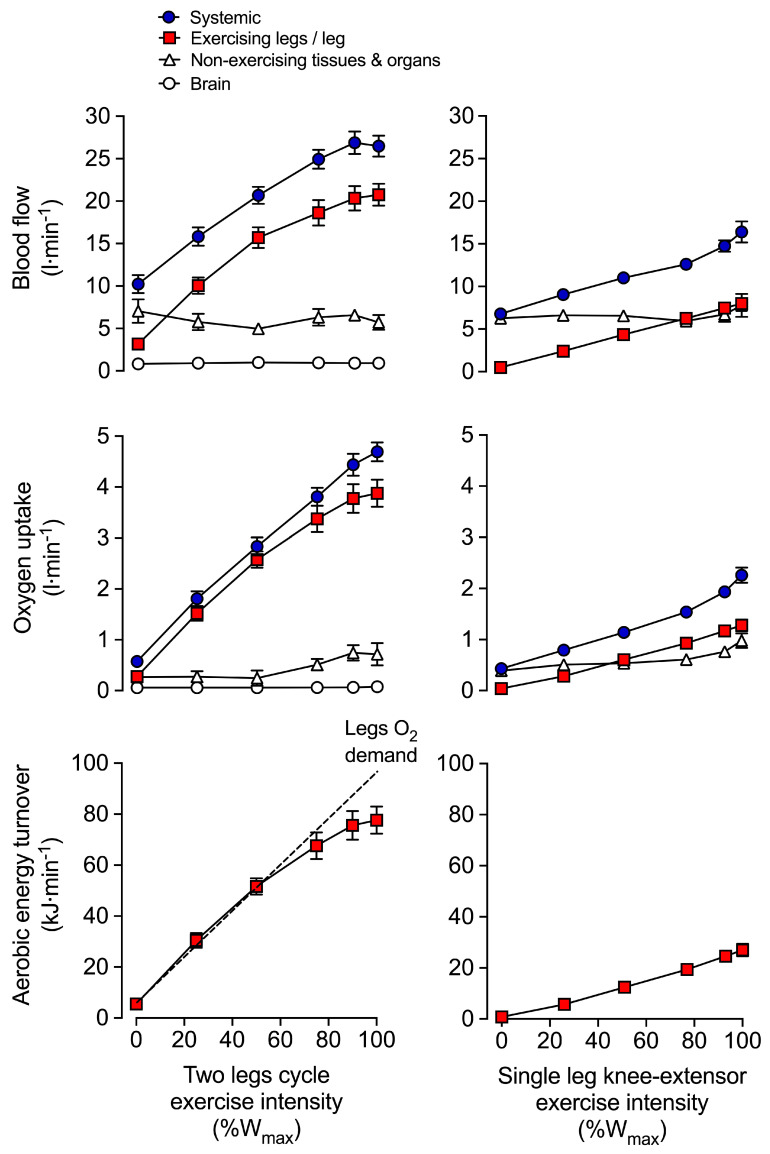
Blood flow, oxygen uptake and energy turnover during exhaustive incremental two-legged cycle exercise and single leg knee-extensor exercise. These data demonstrate that the rate of rise in exercising muscle blood flow, oxygen uptake (*V*O_2_) and aerobic energy turnover is blunted during heavy-to-severe intensity cycle exercise but not during exhaustive single leg knee-extensor exercise. The dotted line indicates the theoretical two legs O_2_ demand, assuming a linear relationship between exercise intensity and *V*O_2_ across all cycle exercise intensities, as found during single leg knee-extensor exercise. Limited data suggest that brain *V*O_2_ is preserved (via increases in O_2_ and substrate extraction from the circulation) in the face of reduced brain blood flow and O_2_ supply during maximal aerobic exercise. Redrawn from data (means ± SE) reported by Mortensen et al. [64], Trangmar et al. [36], Sato et al. [34], and González-Alonso et al. [65].

**Figure 2 cells-11-00383-f002:**
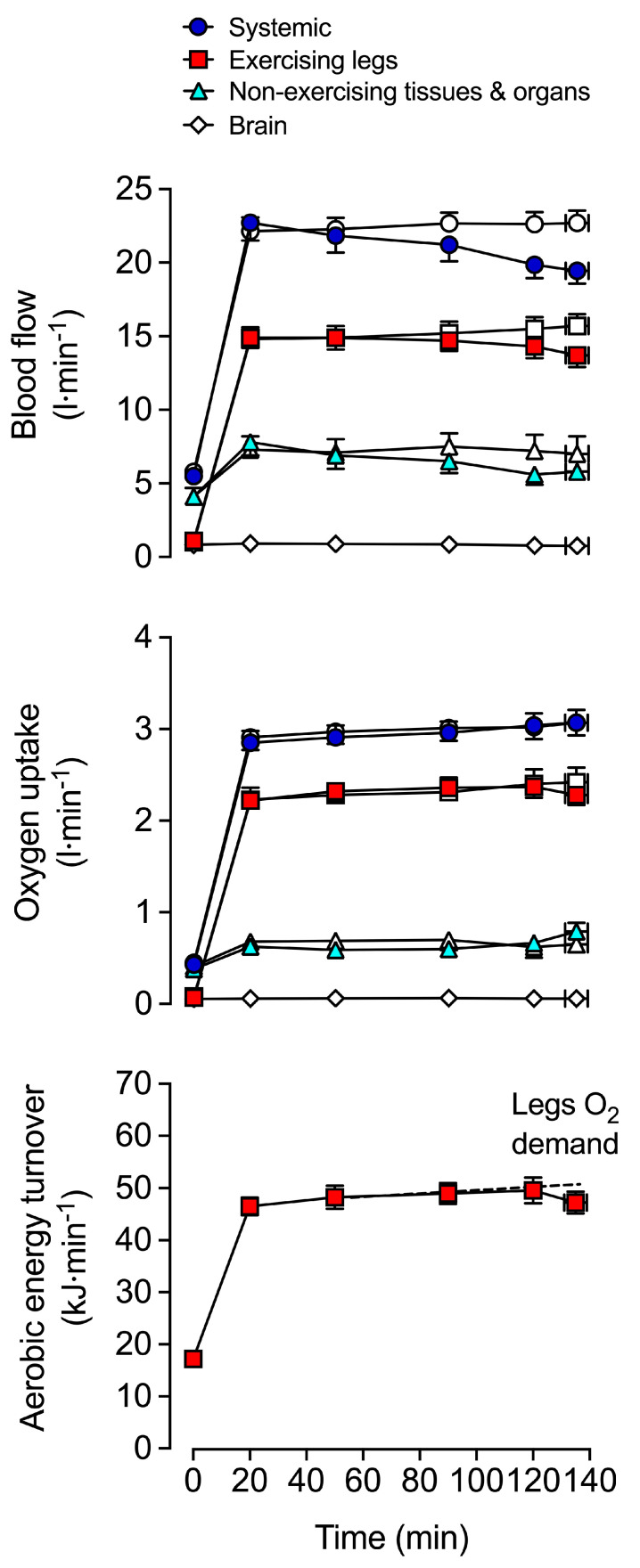
Effects of progressive dehydration and hyperthermia on blood flow, oxygen uptake and energy turnover during submaximal constant load cycle exercise compared to the control euhydration condition. Systemic and regional blood flow progressively decline, whilst oxygen uptake (*V*O_2_) and legs aerobic energy turnover are maintained during prolonged submaximal cycling (~60% *V*O_2max_) with dehydration and hyperthermia to exhaustion compared to control euhydration (corresponding blank symbols) in trained individuals. The dotted line indicates the legs O_2_ consumption (demand) in the control trial. Adapted from González-Alonso et al. [6].

**Figure 3 cells-11-00383-f003:**
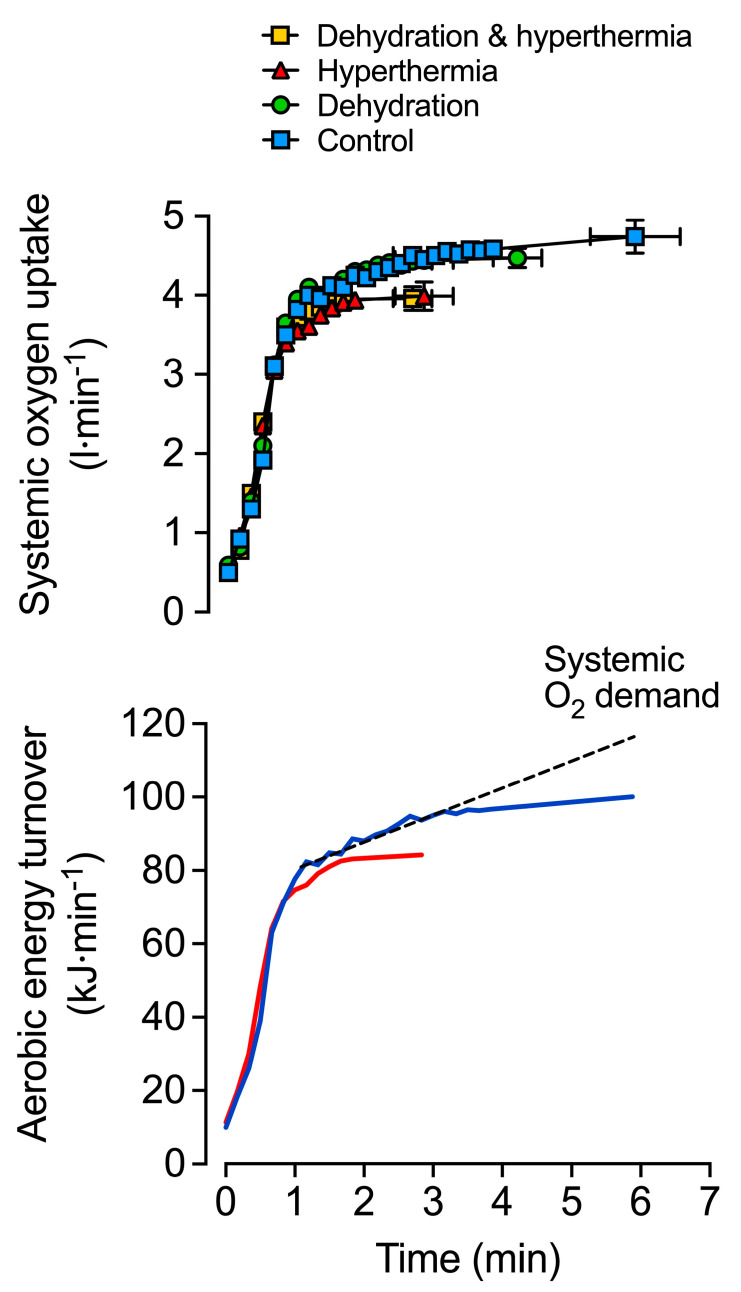
Impact of dehydration and hyperthermia on maximal oxygen uptake and aerobic energy turnover. Oxygen consumption dynamics were measured during constant load maximal cycling (402 ± 4 W) under control, dehydration (4% body weight loss without hyperthermia), hyperthermia (+1 °C and +6 °C increase in T_c_ and skin temperature, respectively) and combined dehydration and hyperthermia. Note that both combined dehydration and hyperthermia and hyperthermia alone impaired *V*O_2max_ and exercise performance by 16% and 51–53% compared to control, without altering the initial absolute *V*O_2_ responses. Preventing hyperthermia in dehydrated individuals restored *V*O_2__max_ and exercise performance by 65% and 50%, respectively. These data demonstrate that aerobic metabolism and maximal endurance capacity can be drastically compromised in the dehydrated and hyperthermic human. The dotted line is the theoretical systemic O_2_ demand. Redrawn from data (means ± SE) reported by Nybo et al. [67].

**Figure 4 cells-11-00383-f004:**
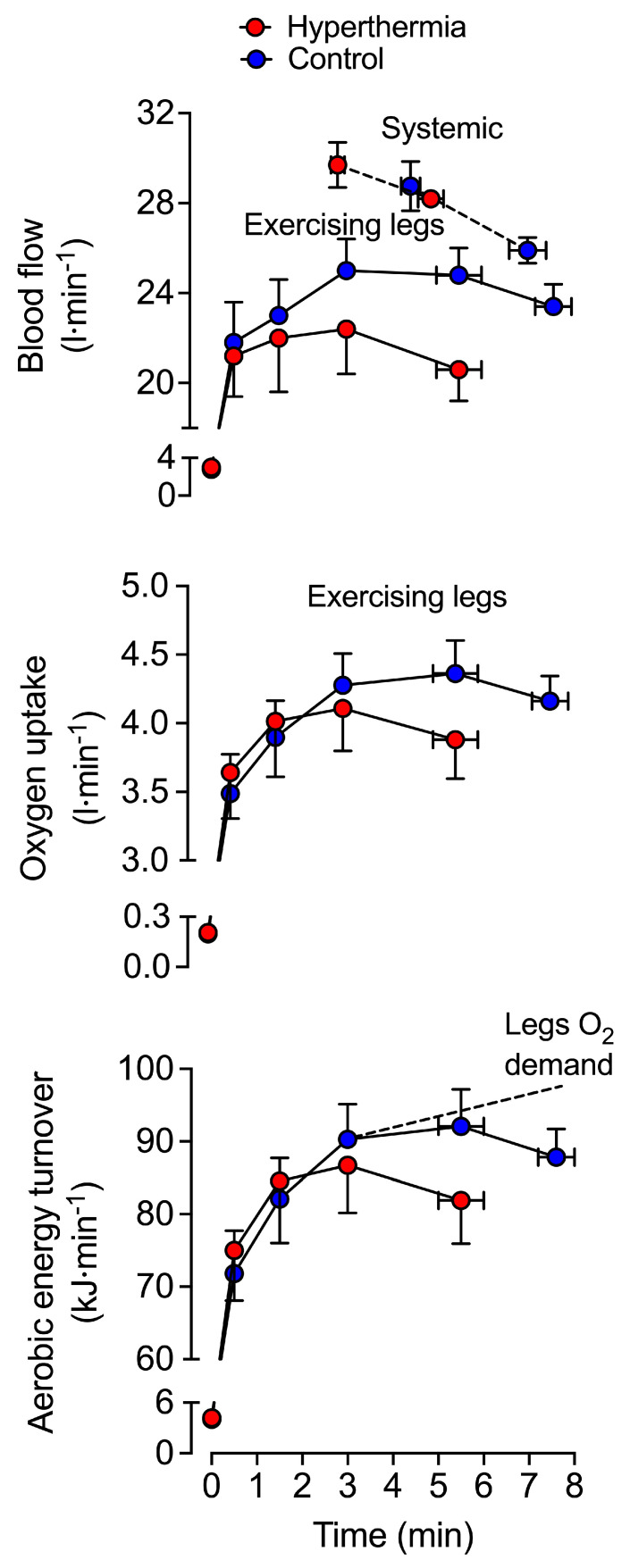
Impact of whole-body hyperthermia on blood flow, oxygen uptake and energy turnover during constant load maximal cycle exercise. Exercising legs (data with continuous lines) and systemic blood flow (data with dotted lines), exercising legs oxygen consumption (*V*O_2_) and aerobic energy turnover responses during constant load cycling (~360 W) to exhaustion under (1) systemic hyperthermia and (2) control (normothermic) conditions. The dotted line depicts the theoretical leg O_2_ demand. Redrawn from (means ± SE) by González-Alonso and Calbet [150].

**Figure 5 cells-11-00383-f005:**
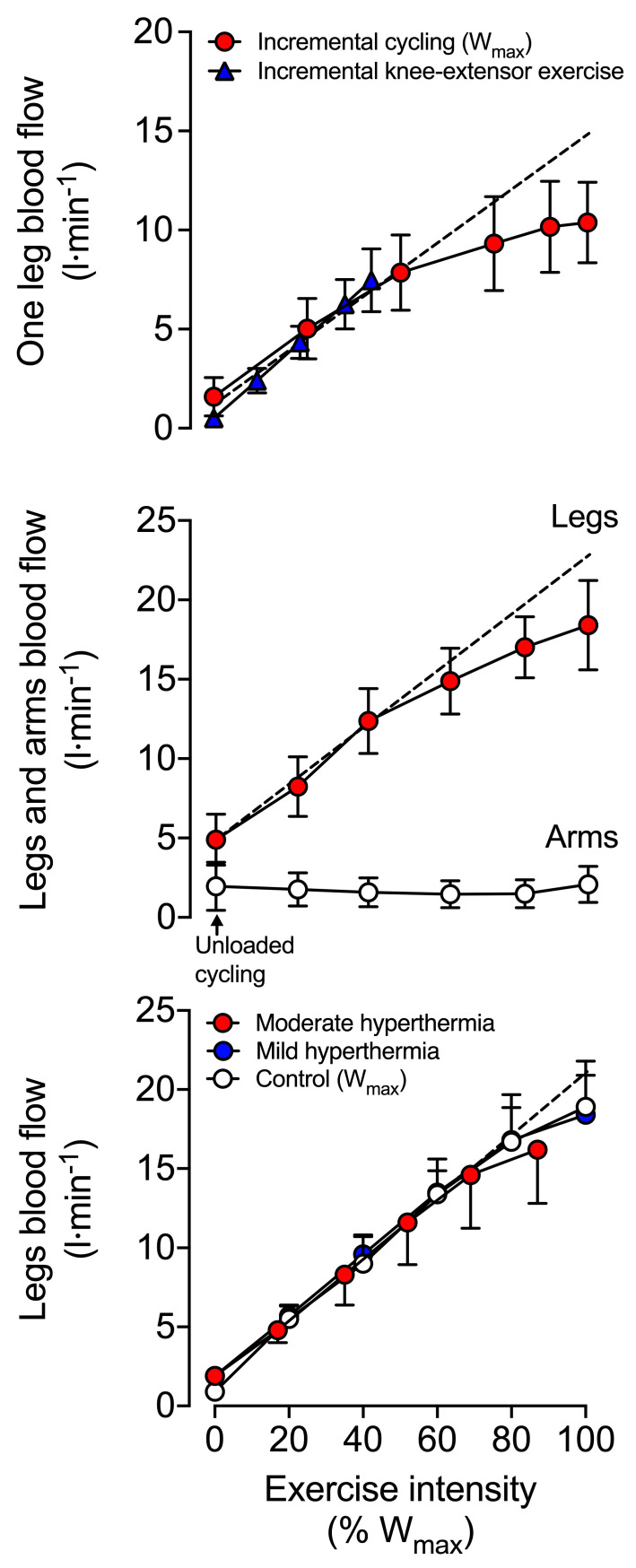
Exercising and non-exercising limb blood flow during incremental exercise to volitional exhaustion. This figure illustrates the attenuation in the rate of rise in locomotor limb blood flow during intense cycling but not during single leg knee-extensor exercise. Blood flow in the non-exercising arms remains largely unchanged during incremental cycling despite the increasing *Q*. The hyperthermia-induced elevation in baseline leg blood flow disappears during whole-body exercise, but an accelerated attenuation in the rate of rise is apparent before exhaustion compared to mild hyperthermia and control conditions. The present graphs were redrawn from data (means ± SD) reported by Mortensen et al. [64], González-Alonso et al. [65], González-Alonso et al. [225] and Trangmar et al. [35].

**Figure 6 cells-11-00383-f006:**
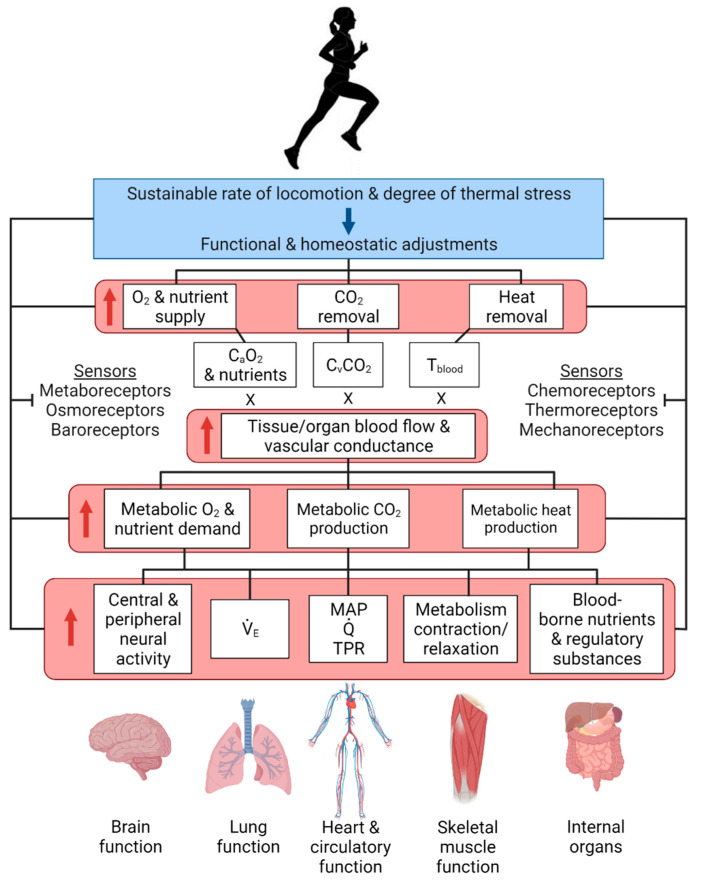
Integrative view of physiological systems during exercise and thermal stress. Increases in the rate of locomotion and the degree of thermal stress induce multiple functional and homeostatic adjustments. These adjustments primarily involve: elevations in O_2_ demand and utilisation, and CO_2_ and heat production in the working skeletal muscles. The associated rise in muscle blood flow ensures appropriate O_2_ supply and CO_2_ and heat removal to the surrounding environment via the circulation. These peripheral responses are linked to the increase in central neuromotor drive, pulmonary ventilation, systemic blood flow and arterial pressure, and are underpinned by elevations in central and skeletal muscle neural activity, as well as global respiratory, vascular and cardiac function. Cellular homeostasis is preserved across all organ systems when the physiological demand of exercise and thermal stress is substantially below the body’s functional capacity. Created with BioRender.com.

**Figure 7 cells-11-00383-f007:**
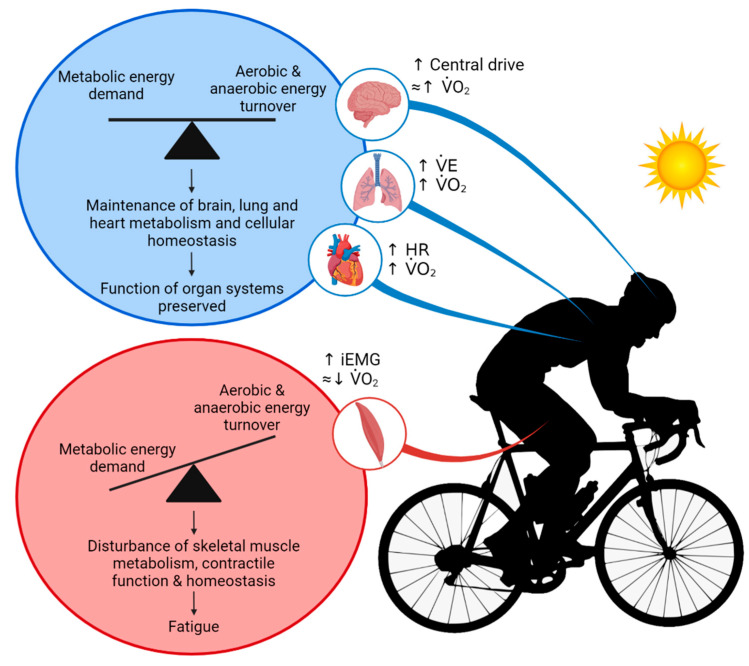
Physiological function and cellular homeostasis during fatiguing exercise. This figure illustrates a theoretical model of the physiological responses to maximal aerobic exercise, with and without environmental heat stress. While the available evidence suggests that compensatory adjustments preserve function and cellular dynamic homeostasis in the brain, the lungs and the heart, a mismatch between energy demand and supply in contracting skeletal muscle leads to compromised local metabolism, suppressed muscle contractile function, impaired force production and ultimately an inability to maintain speed of locomotion or power output, the hallmark of fatigue. Note in Figure 1 and Figure 5 that the rate of rise in exercising legs blood flow is blunted above moderate exercise intensities during incremental two-legged cycling, but not during incremental single leg knee-extensor exercise. The addition of moderate-to-severe whole-body hyperthermia, or combined dehydration and hyperthermia during constant load and incremental exercise (Figure 2, Figure 3, Figure 4 and Figure 5) induce a faster attenuation in the rate of rise or a decline in exercising legs blood flow, leading to disruption in myocyte dynamic homeostasis and accelerated fatigue. In contrast, skin hyperthermia alone—induced via a brief heat exposure using a water-perfused suit—does not speed up fatigue compared to control (Figure 5). Importantly, as illustrated by the exercising legs blood flow in Figure 1, alterations in physiological function and compromised locomotor muscle metabolism can also occur during maximal aerobic exercise in normothermic conditions, underpinned by restrictions in exercising limb blood flow and diminished O_2_ supply. Created with BioRender.com.

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
