# Peer review of "Physiological Function during Exercise and Environmental Stress in Humans—An Integrative View of Body Systems and Homeostasis"

_cells, 2022, doi:10.3390/cells11030383_

Round 1

Reviewer 1 Report

Of the manuscript "Physiological function during exercise and environmental stress in humans - an integrative view of body systems and homeostasis" I report the following:

The manuscript is very well written and addresses the issue raised with excellence. But I have two comments: the text is very long, it presents in some parts a detailed description of each topic, with long sentences that could be shortened. Besides, at the beginning of each organ / system analyzed, a description of its functions is made, perhaps unnecessary, and more suitable for readers who are approaching the physiology of the systems.

A second comment refers to the "control mechanisms" of the homeostatic phenomenon. Here the review is extensive the nervous system, but it is more focused on blood flow and brain metabolism, although in the seventh point (“Organ systems interactions - future research directions”) it addresses some neural control mechanisms itself, Also, it partly touches on autonomic regulation in the cardiovascular chapter but no comment or description is made about the role of hormones as controllers of the internal environment, I believe that a paragraph on this should be incorporated that could be in the mentioned seventh point.

Author Response

Please find our response to all reviewer comments attached.

Reviewer 2 Report

This review is interesting and useful since it examines the body function and the dynamic homeostasis during exercise and environmental stress, from several perspectives. However, nothing is mentioned about the inducible factor of hypoxia as a possible mechanism that would regulate the oxygen demand by the tissues, I think this issue is also relevant. 

Author Response

Please find our responses to all reviewer comments in the file attached.

Reviewer 3 Report

This Review is good and focus interesting.

Author Response

We thank the reviewer for their kind words regarding the manuscript. We hope this will be a valuable teaching resource as well as facilitate future research in integrative physiology.